# A validation framework for neuroimaging software: The case of population receptive fields

**Garikoitz Lerma-Usabiaga**[1,2,3]*, **Noah Benson**[4], **Jonathan Winawer**[4], **Brian A. Wandell**[1,2]

**1** Department of Psychology, Stanford University, Stanford, California, United States of America, **2** Wu Tsai Neurosciences Institute, Stanford University, Stanford, California, United States of America, **3** BCBL, Basque Center on Cognition, Brain and Language, Mikeletegi Pasealekua, Donostia—San Sebastián, Gipuzkoa, Spain, **4** Department of Psychology and Center for Neural Science, New York University, Washington Pl, New York, New York, United States of America

* garikoitz@gmail.com

**Data Availability Statement:** All relevant data are within the manuscript and its Supporting Information files.

## Abstract

Neuroimaging software methods are complex, making it a near certainty that some implementations will contain errors. Modern computational techniques (i.e., public code and data repositories, continuous integration, containerization) enable the reproducibility of the analyses and reduce coding errors, but they do not guarantee the scientific validity of the results. It is difficult, nay impossible, for researchers to check the accuracy of software by reading the source code; ground truth test datasets are needed. Computational reproducibility means providing software so that for the same input anyone obtains the same result, right or wrong. Computational validity means obtaining the right result for the ground-truth test data. We describe a framework for validating and sharing software implementations, and we illustrate its usage with an example application: population receptive field (pRF) methods for functional MRI data. The framework is composed of three main components implemented with containerization methods to guarantee computational reproducibility. In our example pRF application, those components are: (1) synthesis of fMRI time series from ground-truth pRF parameters, (2) implementation of four public pRF analysis tools and standardization of inputs and outputs, and (3) report creation to compare the results with the ground truth parameters. The framework was useful in identifying realistic conditions that lead to imperfect parameter recovery in all four pRF implementations, that would remain undetected using classic validation methods. We provide means to mitigate these problems in future experiments. A computational validation framework supports scientific rigor and creativity, as opposed to the oft-repeated suggestion that investigators rely upon a few agreed upon packages. We hope that the framework will be helpful to validate other critical neuroimaging algorithms, as having a validation framework helps (1) developers to build new software, (2) research scientists to verify the software's accuracy, and (3) reviewers to evaluate the methods used in publications and grants.

**Funding:** Supported by a Marie Sklodowska-Curie (https://ec.europa.eu/programmes/horizon2020/en/h2020-section/marie-sklodowska-curie-actions) grant to G.L.-U. (H2020-MSCA-IF-2017-795807-ReCiModel) and National Institutes of Health (https://www.nih.gov/) grants supporting N.C.B. and J.W. (EY027401, EY027964, MH111417). The funders had no role in study design, data collection and analysis, decision to publish, or preparation of the manuscript.

**Competing interests:** The authors have declared that no competing interests exist.

## Author summary

Computer science provides powerful tools and techniques for implementing and deploying software. These techniques support software collaboration, reduce coding errors and enable reproducibility of the analyses. A further question is whether the software estimates are correct (valid). We describe a framework for validating and sharing software implementations based on ground-truth testing. As an example, we applied the framework to four separate applications that implemented population receptive field (pRF) estimates for functional MRI data. We quantified the validity, and we also documented limitations with these applications. Finally, we provide ways to mitigate these limitations. Implementing a software validation framework along with sharing and reproducibility is an important step for the complex methods used in neuroscience. Validation will help developers to build new software, researchers verify that the results are valid, and reviewers to evaluate the precision of methods in publications and grants.

This is a *PLOS Computational Biology* Methods paper.

## Introduction

Neuroimaging software methods are based on complex architectures, thousands of lines of code, and hundreds of configuration parameters. Consequently, it is a near certainty that some implementations will contain errors. Modern computational techniques (i.e. public code and data repositories, continuous integration, containerization) enable the reproducibility of the analyses and the reduction of coding errors, but do not guarantee the scientific validity of the results.

Computational reproducibility—enabling anyone to obtain the same result for the same input data and code—is one important component of scientific reproducibility [1,2]. Computational generalization means obtaining the same result for the same input data and different software implementations. Computational validity is the further test as to whether the result is correct. For reviewers and scientists alike, it is impossible to establish validity by just reading code.

One expects that any public domain software has been validated in some ways by the developers, and we are sure that this is true for the software we analyze here. We suggest that it is important, however, that the specific validation tests be made explicit to the user. There are many parameters and many experimental conditions in neuroimaging. By making the validation tests explicit, investigators can have a much better understanding of whether the software is likely to be valid for their application.

### Related literature

Validating neuroimaging software has been approached in several different ways. We can consider MRI phantoms as the first validation system. Phantoms, commonly used in quantitative MRI software development, provide ground truth data, [3–5].

There have been other validation efforts in diffusion weighted imaging (DWI) as part of public challenges, such as the Tractometer [6–8], an online tract validation system based on the HCP dataset [9] and a simulated DWI diffusion generated with the tool Fiberbox (http://docs.mitk.org/2014.10/org_mitk_views_fiberfoxview.html).

In addition, investigators have implemented an important part of fMRI validation: software to synthesize realistic fMRI responses data, some for generic task activations [10–13], others more specifically targeted, for example in studies of scotomas [14,15], the impact of eye movements [16], or temporal integration [17]. See here for a review of functional MRI simulations [18].

## Types of validation

The term 'validation' has been used to describe two different types of analyses. The first type, empirical validation, analyzes the reliability of measurements across repeated measures. A high degree of reliability is sometimes thought of as a validation of the method. There are several publications studying the reliability of the pRF measurements [19–24]. Using goodness of fit, test-retest, replication or stimulus and task variation designs, these studies provide important insights into reliability of the measurements.

The second type, the one we implement here, is software validation. This type of validation helps us understand whether the software accurately recovers the ground-truth parameters in a variety of different conditions. As we explain below, by using the validation framework to study pRF software, we learned that the array of pRF tools have a large dependency on the HRF model used in the analysis tool. In typical experimental protocols, the parameter estimates are incorrect unless the empirical HRF matches the HRF used in the tool, a limitation that is not revealed by empirical validation since a result can be repeatable but wrong. This is one of several reasons we advocate for developing software validation frameworks for all important neuroimaging software tools.

## The validation framework

Here, we describe a computational framework for validating and sharing software implementations (Fig 1). The framework is divided into three parts. The x-Synthesize part comprises methods to produce synthetic test data with known parameters in a defined file format. The x-Analyze part uses the test data as inputs to the algorithms under test. These algorithms are incorporated in containers that accept test data inputs and produce output files in a well-defined format. The x-Report tools compare the outputs with the ground-truth parameters in the x-Synthesize part. The x-Analyze containers can also analyze experimental rather than synthetic data placed in the input file format. This framework permits the user to compare multiple algorithms in different containers.

We use 'x' as a placeholder that is replaced by the name of a specific analysis. The framework and methods can be extended to many neuroimaging tools. In this paper, we

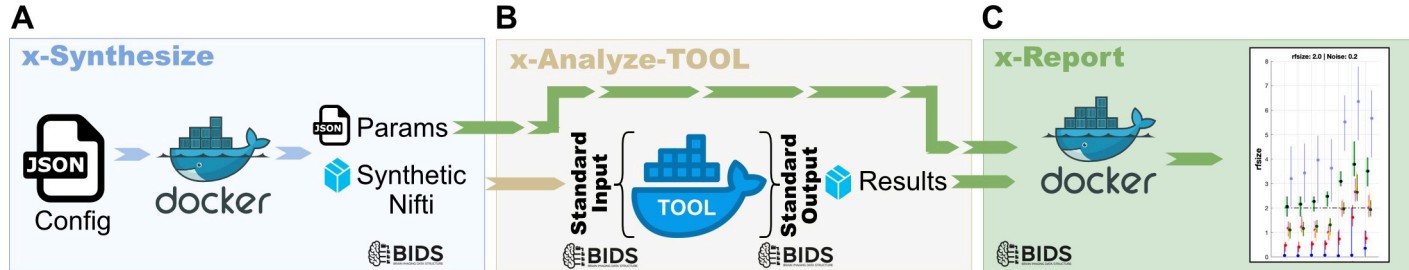

**Fig 1. Validation framework overview.** The framework is composed of three main components. *x-Synthesize*: synthesis of neuroimaging data from ground truth parameters; *x-Analyze-TOOL*: implementation of analysis tools and standardization of inputs and outputs; *x-Report* compares the tool outputs the ground truth parameters. The three components are implemented within containers. Container parameters are specified by JSON text files.

implemented population receptive field (pRF) methods for functional MRI data. The main motivation of beginning with pRF methods is the possibility of accurately generating synthetic data because the pRF is a model. Hence, we can ask how accurately the software recovers the model parameters. Following the guidelines, the pRF validity framework is composed of three main components: (1) synthesis of fMRI time series from ground truth pRF parameters, (2) implementation of four pRF analysis tools and standardization of inputs and outputs, and (3) tool-independent report creation to compare the results with the ground truth parameters. The different components are implemented using containerization methods to guarantee computational reproducibility. All components and testing datasets are publicly available. A user can run the framework with only Docker and a text editor installed.

## Methods

The software we developed comprises two main components: the open-source code repository (https://github.com/vistalab/prfmodel) and the containers (Docker/Singularity). The containers we implemented for the population receptive field analysis can be installed with one command ('docker pull'). Users who prefer Singularity can convert the containers from Docker to Singularity (https://github.com/singularityhub/docker2singularity). The input/output of every element of the framework follows the Brain Imaging Data Structure (BIDS) format [25]. This means that x-Synthesize will generate synthetic data in BIDS format, and that the output of the x-Analyze and x-Report will be in BIDS derivatives format.

The following sections describe the three parts of the framework. The case we describe tests population receptive field (pRF) algorithms: mrVista (https://github.com/vistalab/vistasoft), AFNI (https://github.com/afni/afni), Popeye (https://github.com/kdesimone/popeye), analyzePRF (https://github.com/kendrickkay/analyzePRF). These four are a subset of the public implementations (https://github.com/vistalab/PRFmodel/wiki/Public-pRF-implementations). It is our intention for the framework to be applicable to many other analyses.

## Synthesize

The prf-Synthesize container generates synthetic BOLD time series data. We use the naming convention x-Synthesize where 'x' is replaced by the algorithm under test. For example, glm-Synthesize, dti-Synthesize, csd-Synthesize, and so forth. See S1 File for a basic *How to use* guide.

The synthetic BOLD signal is created using the forward model that is implicit when searching for pRF parameters. The forward model comprises several components (Fig 2). The stimulus is represented by a sequence of 2D binary images that represent the presence (1) or absence (0) of contrast at each location. The two-dimensional receptive field (RF) is also represented as a matrix; the inner product of the stimulus and the RF matrix produces a response time series.

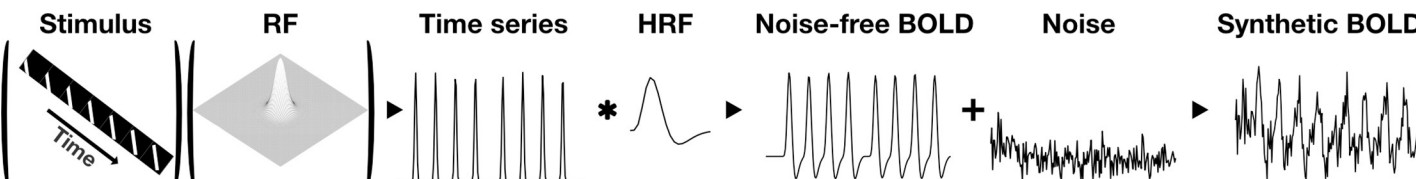

**Fig 2. prf-Synthesize: Simulation of ground-truth BOLD data with a forward model.** The prf-Synthesize container reads the JSON parameter file to define the RF parameters, and a NIfTI file to specify the stimulus contrast time series. The noise-free BOLD signal is calculated from the inner product of the stimulus and RF, which generates a time series. This time series is convolved with the HRF to obtain the noise-free BOLD signal. A parameterized noise model produces a signal that is added to generate the synthetic BOLD signal. *RF: Receptive field; HRF: Hemodynamic response function. BOLD: blood-oxygen-level dependent.*

The time series is convolved with the hemodynamic response function (HRF) to obtain the noise-free BOLD signal. A noise model produces a signal that is added, generating the synhtetic BOLD signal.

Each of these components is controlled by explicit parameters that are available through the prf-Synthesize interface. The outputs at each stage of the calculation can be visualized and modified if the user chooses to install the Matlab code (see Supporting Information for details).

**Receptive Field.** The population receptive field is represented as a 2D function over the input image (stimulus). The simplest and most widely used pRF models are implemented in prf-Synthesize (Fig 2-RF). The Gaussian receptive field has five parameters:

$$G_i(x, y, \sigma_1, \sigma_2, \theta) \qquad (1)$$

- The center position of the receptive field (x, y).

- The standard deviations ($\sigma_i$) of the two axes of the ellipse

- The angle ($\theta$) of the main axis (larger sigma). (Zero for a circular receptive field, $\sigma_1 = \sigma_2$)

In addition to the Gaussian, we implemented a Difference of Gaussians (DoG) model. In this case, there is also a relative amplitude for the center and surround Gaussians

$$DoG = G_c(x, y, \sigma_1, \sigma_2, \theta) - aG_s(x, y, \sigma_1, \sigma_2, \theta) \qquad (2)$$

The validation tests in this paper were performed only for the circular, single Gaussian model ($\sigma_1 = \sigma_2, \theta = 0$). The code is more general and in the future we plan to use this framework to test other receptive field models (anisotropic, difference-of-Gaussian).

**Stimulus.** We simulate visual stimuli using the methods described on this web page (https://kendrickkay.net/analyzePRF/). That code generates a binary stimulus that represents the locations where there is non-zero stimulus contrast. The default stimulus represents 20 deg of visual angle and is stored as a (row, col, time) tensor with 101 row and column samples and 200 temporal samples. The stimulus parameters can be adjusted using the entries in the JSON file that is the input to prf-Synthesize. The default stimulus is a bar that sweeps through the visual field in multiple directions and includes some blank periods. The default timing parameters are set to a 200 sec presentation with a sampling of 1 sec.

**Hemodynamic Response Function (HRF).** The default HRF is the sum of two Gamma probability distribution functions,

$$x^{(a-1)}\exp(-\beta x) \qquad (3)$$

Each Gamma distribution has two parameters, and a third parameter defines the relative scaling. As we show below, it is quite significant that different tools implement different HRFs, either in their parameters or in their functional form. We explore the impact of the HRF extensively below.

**Noise models and parameter selection strategies.** We implemented three types of additive noise: white noise, physiological noise (cardiac and respiratory) and low frequency drift noise [10,11,18,26,27]. The level of the white noise is controlled by a single parameter that sets the amplitude. Respiratory, cardiac and low-frequency noises are each controlled by four parameters: frequency, amplitude and a jitter value for frequency and amplitude. The jitter value modifies the frequency and amplitude so that each time the noise model is run the parameters values are randomized.

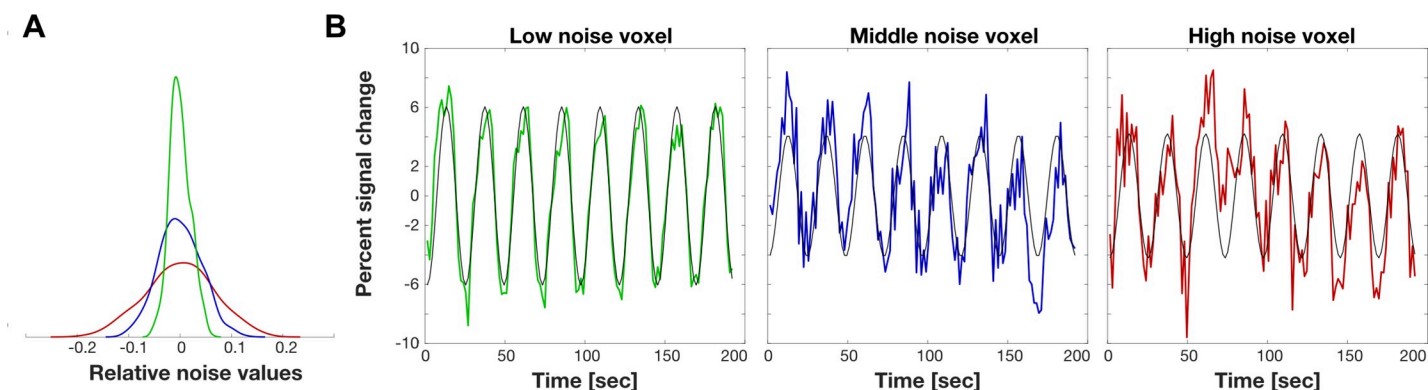

**Fig 3. Noise parameter estimation.** We analyzed BOLD data collected with an on-off visual contrast stimulus (8 cycles/run, four repetitions). Noise was assessed by comparing the BOLD time series with a harmonic at the same frequency. A) The probability density functions of the noise values of three voxels with low (green), mid (blue) and high (red) noise. B) The colored curves are the mean BOLD time-series plotted as percent modulations (mean of the four repetitions). The black curves are the eight cycle harmonic fit to the data. The noise is the difference between the two. The blue curve is the middle of the noise level found in visual cortex. The other two curves show low and high noise examples that are often found in real measurements. *Low noise*: SNR = 6.55dB; *mid noise*: SNR = -3.03dB; *high noise*: SNR = -6.68dB.

For the analyses below we simulated the BOLD time series at three noise levels (Fig 3). We selected the noise levels by analyzing the measured BOLD time series from a participant who viewed 8 repetitions of a large-field on/off contrast stimulus. The BOLD data were sampled every 1.5 seconds, each cycle comprised 24 seconds (0.042 Hz) and consisted of 22 axial slices covering the occipital lobe, with 2.5 mm isotropic voxels. The experiment was repeated four times. Three gray matter voxels were selected and the range of noise levels in these voxels were used to set the range of noise levels in the simulation (the fMRI data are available at https://osf.io/3d8rp/).

These three reference voxels were selected using an automated procedure. First, grey matter voxels were identified whose mean BOLD signal exceeded 75% of the total mean BOLD signal. Second, from this set we selected voxels with high local coherence values. The coherence at each voxel was calculated by dividing the amplitude at the stimulus frequency by the mean amplitude of the frequencies in a window around the stimulus frequency. The amplitude spectrum of the BOLD signal of a voxel with high coherence has a local peak at the stimulus frequency, 0.042 Hz. The amplitude at the stimulus frequency in this example is about 10x higher than the expected noise amplitude. The simple on/off stimulus generates significant responses in large regions of the occipital lobe. We selected voxels that responded significantly to the stimulus, at least 80% of the maximum coherence.

Third, the BOLD signal was converted to percent signal change (PSC) by subtracting and dividing each time series by its mean. We selected the voxels with signal percent change between 8% and 12%. The contrast was calculated as the mean between the maximum and minimum values of the time series. Fourth, noise was measured in the surviving voxels as the difference between the BOLD signal and an 8 cycle sinusoidal fitted to the signal. We selected 3 voxels based on their noise distributions, at 95%, 45% and 10% (Fig 3A) of the voxel with the smallest noise level. Fig 3B shows the mean value of the 3 selected voxel time series and the corresponding fitted sinusoidal. Signal to noise ratio (SNR) was calculated by the ratio of the root mean squared error of the signal (the fitted sinusoidal) and the noise (time series minus the fitted sinusoidal).

Based on these values, we found three parameter settings that correspond to the low- mid— and high-noise levels in the prf-Synthesize tool (Fig 4). These parameters can be set by the user as well. The levels shown in the figure correspond to a single acquisition of the time series. It is

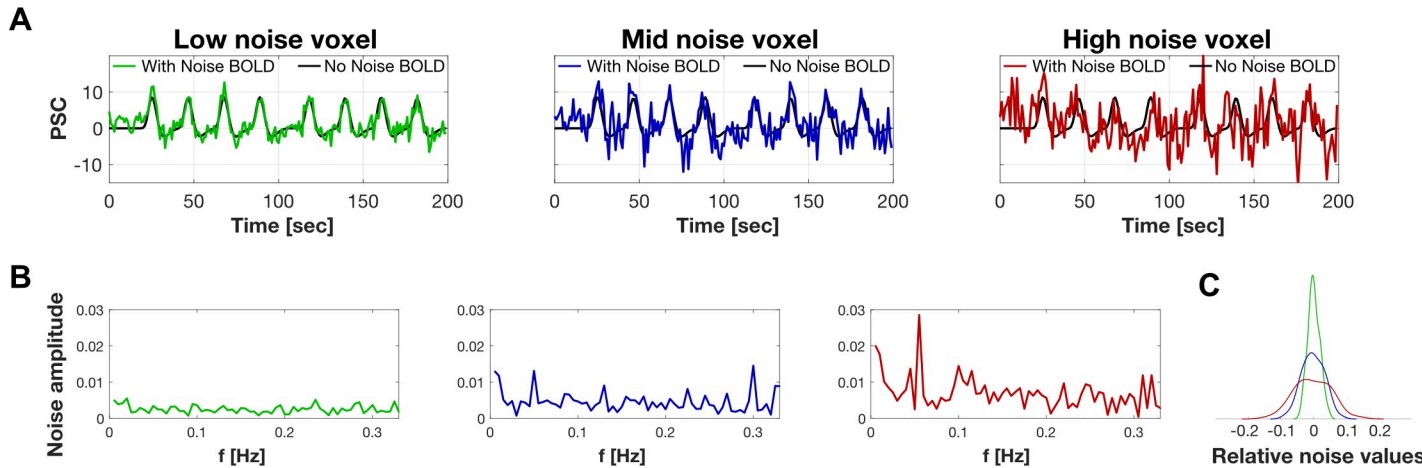

**Fig 4. prf-Synthesize BOLD signals calculated for a moving bar retinotopy stimulus.** (A) The model calculation of a BOLD signal to a typical moving bar stimulus under the three noise levels. (B) Noise amplitude spectrum for each of the cases in A. (C) The probability density function of the noise values. The simulations match the noise distributions in Fig 3A. *PSC: percent signal change; f: frequency. Low noise: SNR = 5.29dB; mid noise: SNR = -0.51dB; high noise: SNR = -4.29dB.*

common practice to average several scans to increase SNR. For example, averaging 3 mid-noise level time series increases the SNR 2–3 dB.

## Implementation: Analyze

The prf-Analyze containers implement the pRF tools. The inputs to the container are defined using standard file formats and directory organization. The software repository includes a base container specification so that a pRF tool developer can insert their code and test the implementation using our test framework. To this point, we have implemented four different prf-Analyze containers:

1. **prf-Analyze-vista:** the vistasoft's pRF Matlab implementation, heavily based on a graphical user interface.

2. **prf-Analyze-afni:** AFNI's command line based C++ pRF implementation.

3. **prf-Analyze-aprf:** analyzePRF is Kendrick Kay's PRF Matlab implementation.

4. **prf-Analyze-pop:** Popeye is a pRF python implementation.

Each container takes the stimulus file and the BOLD time series represented as NIfTI files and organized using the BIDS file naming convention and directory tree [25]. This is the format that is generated by prf-Synthesize but could just as well be data from an empirical study. The prf-Synthesize tools are designed to work with BOLD data following general pre-processing, including: removal of the initial volumes to allow longitudinal magnetization to reach steady-state; correction of differences in slice acquisition times; correction for spatial distortion; motion correction. See the Supporting Information for more information about the prf-Validation User's Guide.

In the course of implementing and validating the containers, we identified a few issues that needed to be corrected with the original downloads. These issues were reported to the developers and they have updated the code. We note only the issues that might impact results run using the versions in use prior to mid 2019. Three main issues were related to specification of the temporal sampling of the HRF (https://github.com/kdesimone/popeye/pull/64) and the normalization of input BOLD signals (https://github.com/kdesimone/popeye/pull/62) in

popeye, and an error in the formula of the ellipse in AFNI ([https://afni.nimh.nih.gov/pub/dist/doc/misc/history/afni_hist_BUG_FIX.html](https://afni.nimh.nih.gov/pub/dist/doc/misc/history/afni_hist_BUG_FIX.html)). An extension was added to mrVista to enable using synthetic data comprised of one-dimensional NIfTI data format.

### Report

The third component of the framework, prf-Report container, compares the prf-Analyze-tool outputs with the parameters that specified the ground-truth values used by prf-Synthesize. The input to prf-Report report is a json file that specifies the location of the ground-truth and analyzed data files. The prf-Report container reads these files and creates a summary file that includes the key ground-truth parameters and the corresponding prf-Analyze estimates. The container also generates summary plots that assess algorithm performance. These plots are stored in scalable vector graphics (SVG) format and images in portable network graphics (PNG) format that are readily usable in any report or web-site. See the <u>Supporting Information</u> for the prf-Validation User's Guide.

## Results

We analyzed the different tools using synthetic data with and without noise. The main conclusions are observable in the noiseless analysis and also valid for the noise case. The main value of the noise analysis is to illustrate the change in the size of the confidence intervals of the parameter estimates as noise increases.

### Noiseless analysis

**Accuracy.** As a minimum requirement, we expect that the software will return an accurate estimate of the parameters when the synthetic signal is (a) noise-free, and (b) generated using the same assumptions by the analysis, including a matched HRF. Fig 5 compares the four estimates of a circular population RF located at (3,3) deg and with a size of 2 degree radius.

When first downloaded the algorithm parameters and functionalities differ. For example, some of the algorithms fit both the pRF parameters and the HRF function. Others include a compressive spatial summation by default. Some of the algorithms permit fitting of more complex pRF shapes (non-circular, difference of Gaussians). Before exploring these additional features—and they should be explored—we decided to validate the basic algorithm. The comparisons in Fig 5 were performed with a base case that all of the tools can analyze: a

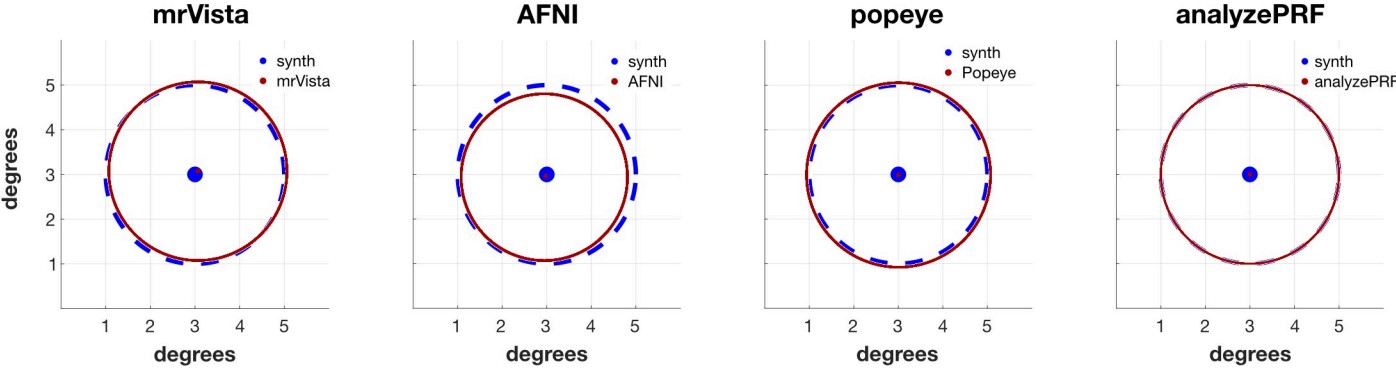

**Fig 5. prf-Report for noise-free position and size estimation.** prf-Synthesize created BOLD signals from a moving bar stimulus and a gaussian RF centered in x = 3 deg, y = 3 deg, with a 1 SD radius of 2 deg. Four prf-Synthesize tools estimated the center and size. In each case the noise-free BOLD signals were calculated using the same HRF implemented in the tool. All tools estimated parameters using the linear pRF model.

[https://doi.org/10.1371/journal.pcbi.1007924.g005](https://doi.org/10.1371/journal.pcbi.1007924.g005)

circular Gaussian pRF, a linear model, and an assumed HRF [28]. This is a simple test that all the tools can fit, but we emphasize that this was not generally the default settings when the software was downloaded.

**HRF mismatch.**   Every prf-Analyze tool assumes a default HRF. The analysis in Fig 5 was constructed so that the HRF in the prf-Synthesize and prf-Analyze tool matched. For that analysis, we generated a different synthetic BOLD signal for each tool. In this section, we study the performance of the tools when the HRF assumption does not match the prf-Synthesize assumption. We generated BOLD time series using four different HRFs that systematically differed in their width (Fig 6A). We used prf-Validation for these different synthetic data. There is a large and systematic bias in the size estimate that depends on the width of the HRF (Fig 6B). There is a correspondence between the width of the HRF used to synthesize the BOLD signal and the size of calculated pRF: if the width of the HRF used to synthesize is smaller than the width of the HRF assumed by the tool, then the tool will underestimate the size of the pRF (and the opposite is true as well). The Boynton HRF [29] was selected for illustration purposes, because the lack of undershoot made the example clearer, but similar effects were observed with all HRF shapes. We observed mismatch in eccentricity and polar angle as well, but at magnitudes of about a tenth of a degree. For this stimulus and the noise-free case, we consider the HRF effect on polar angle and eccentricity to be negligible.

## Noise analysis

We added three levels (low, mid, high) of white, physiological and low drift noise to the original noise-free signals, and we synthesized 100 repetitions per each. Each repetition was different due to the random nature of the white noise and the jitter introduced to the amplitudes

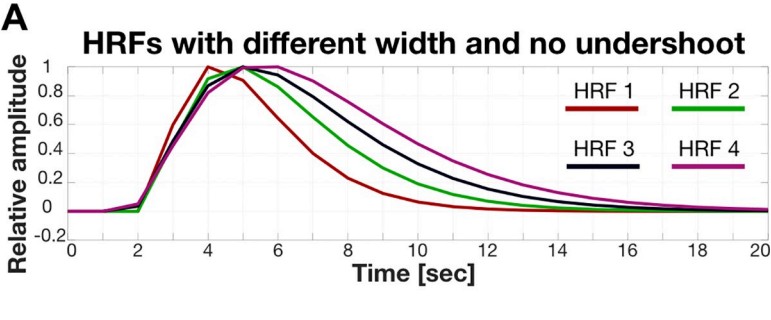

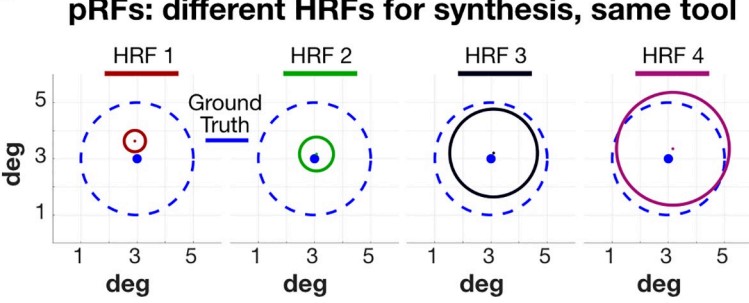

**Fig 6.  RF size dependence on the HRF shape** (A) Four all-positive HRFs with different widths. (B) The ground-truth signals encoded a circular pRF centered at (3, 3) deg with a radius (1 SD of the Gaussian) of 2 deg (dashed, blue circles). The solid colored circles show the estimated pRF calculated with a representative analysis tool (analyzePRF); all prf-Analyze tools have the same pattern. There is a systematic relationship between the HRF width of the estimated pRF radius.

## HRF FOR SYNTHESIS

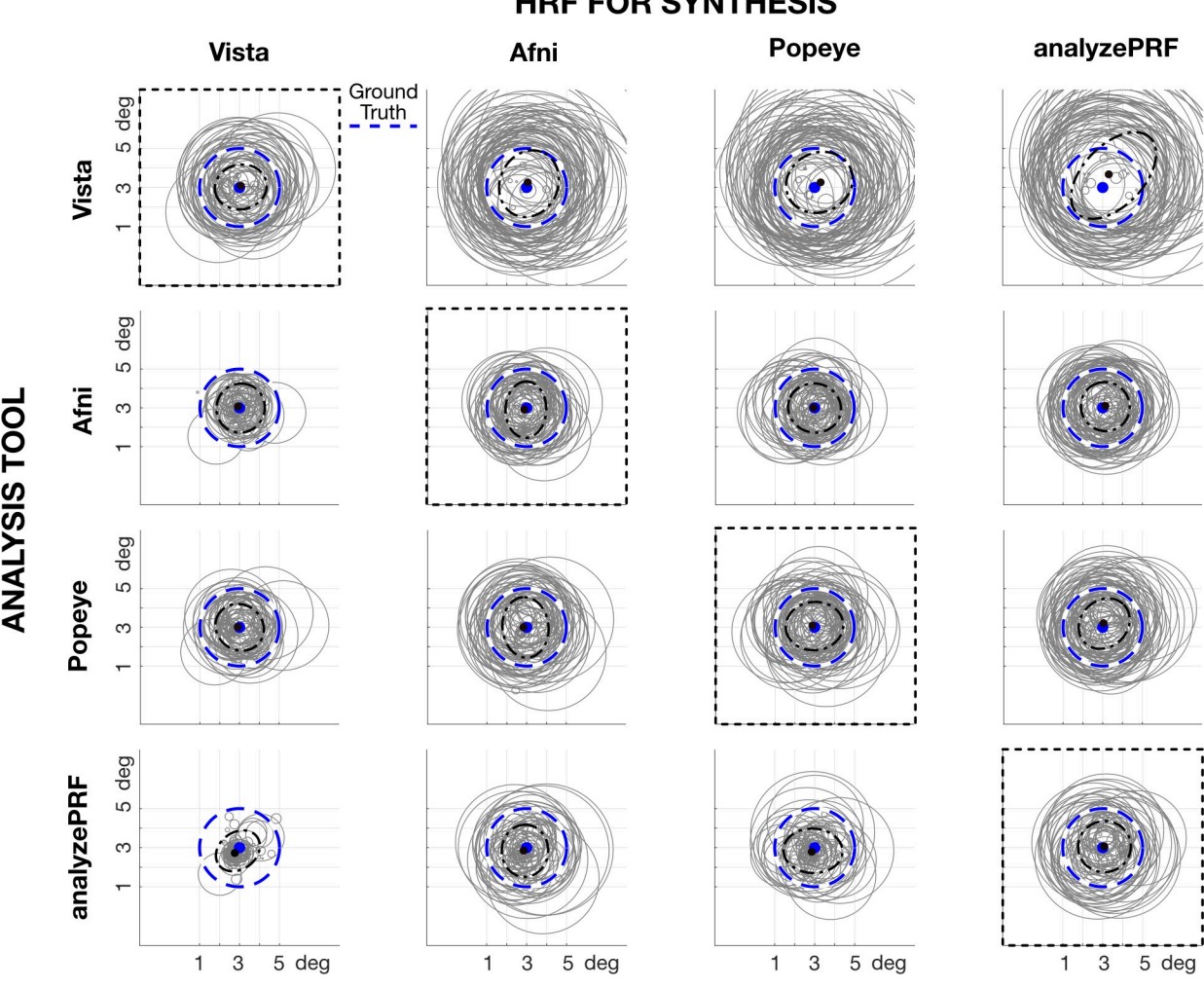

**Fig 7. Noise analysis and HRF dependence.** Each column shows the HRF used in pRF-Synthesize to simulate the BOLD time series. The prf-Synthesize tool created BOLD signals with the typical noise level (mid level, see Fig 4) and a circular RF centered at (3,3) deg and 2 deg radius (dashed blue circle). Each row corresponds to the pRF-Analyze tool, with its default HRF, that was used to analyze the data. The gray circles show the central 90% of the estimated RFs. The central ellipsoid includes 90% of the estimated centers (dashed-dot, black). The central black dot shows the median center location. The HRF used in the synthesis matches the HRF assumed in the analysis in the plots along the diagonal (dashed black rectangles). Above the diagonal, the synthesis HRF is narrower than the analysis HRF; below the diagonal the opposite is true. For other center and radius simulations, see supporting Figures B-E.

and frequencies. In Fig 7, we show the middle noise case, and illustrate the effect of noise on the pRF estimates (jitter within each plot) and the effect of the relation between the HRF assumed in analysis and the HRF used in the synthesis (differences between plots). Here we selected illustrative center and radius values (x = 3, y = 3, radius = 2deg). We tested other combinations of x, y, and radius in supporting Figures B-E. In all cases, we show only the 90% confidence interval of the values, due to the presence of outliers in some of the analysis tools. The centered dash-dot ellipse represents the spread of the centers. The dashed blue curve is ground truth.

In fMRI experiments we have no control over HRFs, which differ between people and across cortical locations. We created a simulation using a range of HRFs for each tool (Fig 8). The confidence interval of the pRF radius is almost ±2 degrees. This value can change depending on the size of the ground truth pRF size, but the general pattern is the same.

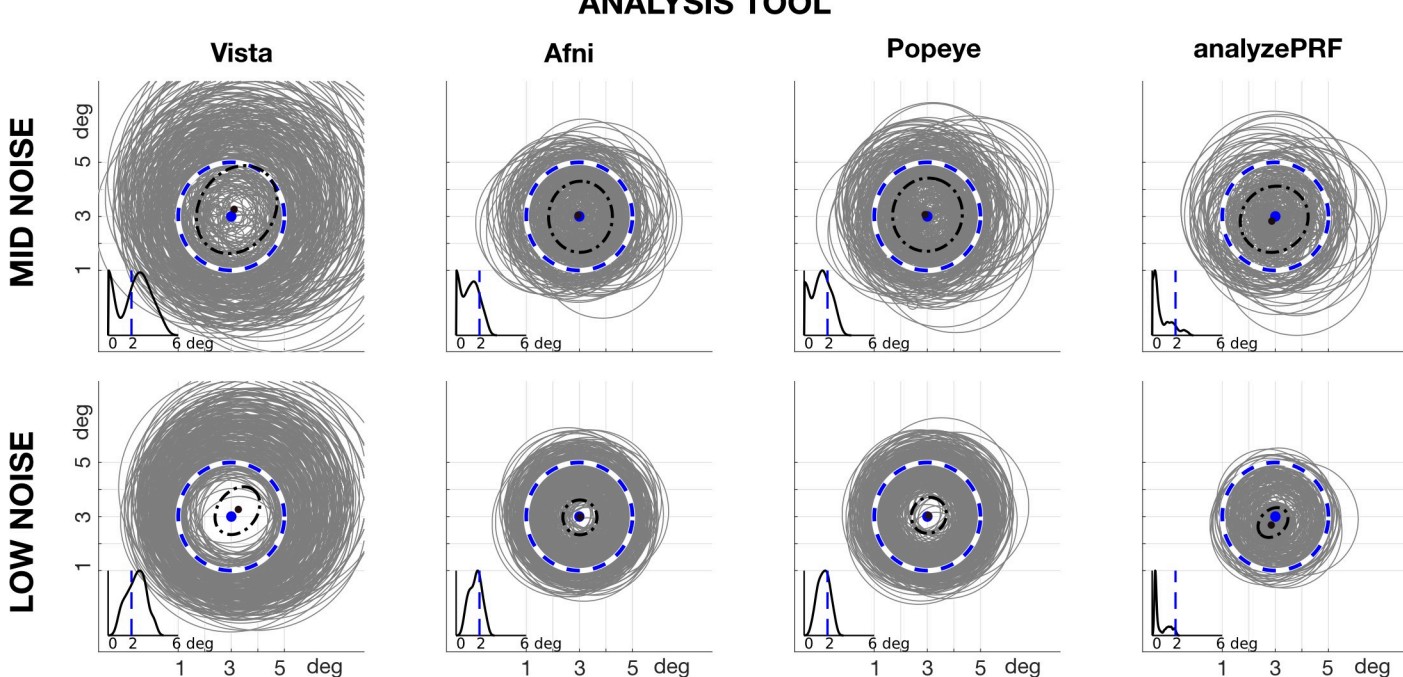

**Fig 8. HRF variation.** The synthesized circular pRF was centered at (3,3) deg with a radius of 2 deg. The data were simulated with prf-Synthesize 400 times, and each time using one of the four prf-Analyze HRFs to synthesize the data. The estimated circular pRFs (gray circles) were calculated using the default HRF implemented in each tool. Outliers are eliminated by showing only fits within the 90th percentile of the pRF size estimates. The inset shows the probability distribution of the pRF sizes, where the blue vertical line shows the ground truth (2 deg).

## HRF dependence

We found that the default HRF assumed in the analysis tool can have a large effect on the size of the estimated pRF. The effect on eccentricity and polar angle was negligible. This is due in part to the stimulus design, which included bar sweeps in both directions along several axes (e.g., left-to-right and right-to-left). If the stimulus were not symmetric (e.g., containing left-to-right sweeps but not right-to-left), then a mismatch between the HRF assumed by the analysis tool and synthesis tool would have resulted in larger errors in the estimated pRF center position.

The HRF impact on pRF size estimates raises two main concerns. The most obvious one is that we cannot compare the absolute size results across tools with different HRF assumptions; the comparison only makes sense within the same analysis tool. The second and more concerning is that even with the same tool, two participants may have different HRFs, or the same person may have different HRFs in different locations in the brain [30]. This HRF dependence has been noted by other investigators who have adopted various means to estimate individual participant HRFs [24].

There may be many approaches to mitigating the HRF dependence. We describe approaches here.

**Computational mitigation.** One approach to reducing the impact of the HRF is to estimate it. There are many ways to approach the HRF estimation. Some investigators allow every temporal sample to be a free parameter [31,32]. Measuring a subject's HRF in a separate experiment adopts this approach. Other investigators parameterize the range of HRF possibilities, using a model with a few parameters (see [30] for a review on alternatives). Some investigators use independent experiments to estimate the HRF and the pRF; others perform a joint

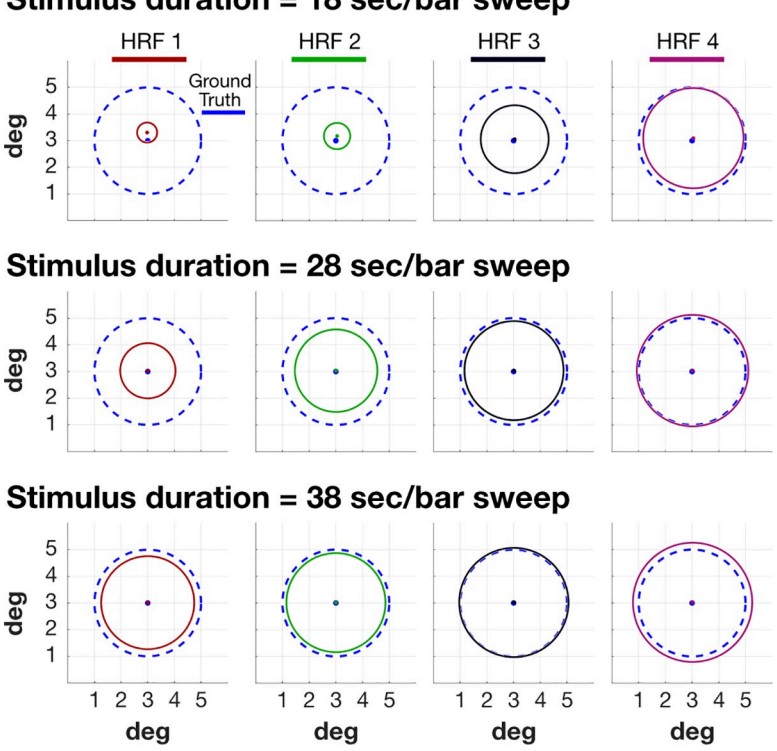

**Fig 9. Stimulus duration reduces the impact of HRF variation.** Slowing the bar movement through the visual field reduces the HRF mis-match impact. The three rows extend the calculation in Fig 6, using three bar speeds: 18, 28 and 38 sec per sweep. The bar sweeps were slowed by changing the total duration of the stimulus. For the same TR of 1s, the same 8 bar sweeps are shown to the subject, effectively doubling the sample points between the fastest and slowest options. The same results were observed when using TR = 2s for the slowest bar sweep. Other details as in Fig 6.

estimation. These alternatives recognize the possibility that the HRF is stimulus dependent, in which case the estimate from an impulse may not generalize to the estimate from the retinotopic data.

Accurately estimating the HRF in every voxel is challenging and vulnerable to noise. Two of the prf-Analyze tools, mrVista and Popeye, have an option to fit the HRF parameters in addition to the pRF parameters. Preliminary validation tests suggest that the effectiveness of the HRF fitting option, although positive, was small. This is a good direction for future exploration.

**Empirical mitigation.**   A second approach to reducing the impact of the HRF is to change the stimulus. Slowing the bar sweep across the screen reduces the impact of the HRF on the estimated pRF size (Fig 9). The three rates shown in the figure are representative of values found in recently published measurements [33–35]. When the bar sweep is slowest (38 sec), the HRF mismatch has much less effect on the estimated pRF size. Selecting this approach substantially increases the amount of time required to obtain the measurements.

Randomizing the position of the bar, rather than sweeping it across the visual field, also reduces the impact of the HRF on the estimated pRF size (Fig 10A). The benefit of position-randomization depends on the specific randomization. It may be possible to find a specific randomized pattern that optimizes the validity of the pRF size estimate for a particular HRF and pRF size.

Choosing a randomized bar position, rather than a sweep pattern, comes at the cost of a significant loss of contrast in the BOLD response (lower signal-to-noise, Fig 10B). For the sweep

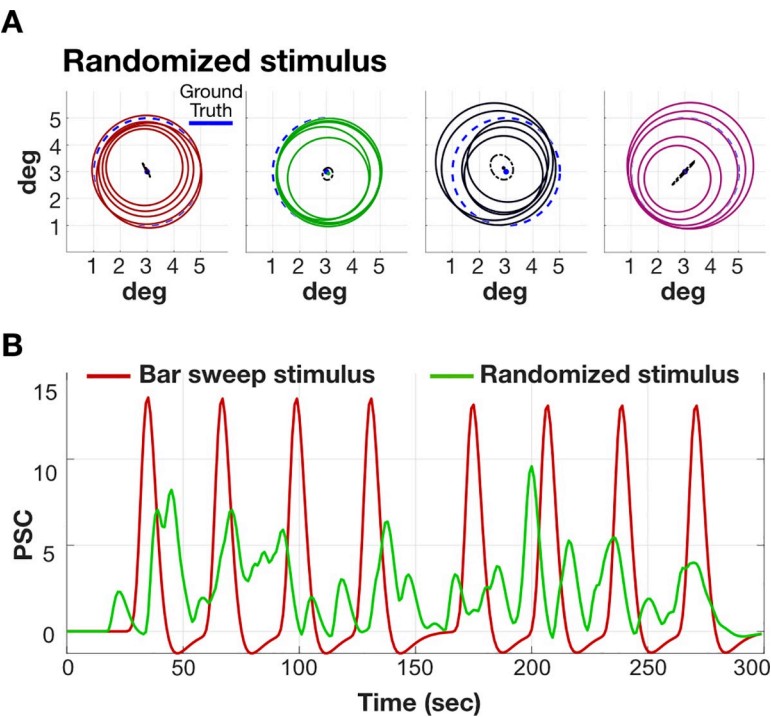

**Fig 10. Stimulus randomization reduces the impact of HRF variation.** (A) Randomizing the bar positions in time reduces the HRF mis-match impact. This figure extends the calculation in Figs 6 and 9, using 5 different randomization seeds. Other details as in Fig 6. (B) Noise-free time series corresponding to bar-sweep stimulus (red) and randomized stimulus (green). The SNR will be lower due to the randomization of the stimuli.

patterns the bar systematically passes through the pRF of each voxel, driving the response to its maximum due to temporal summation from the HRF. For the randomized positions, the bar is positioned over the pRF only briefly, consequently reducing the response contrast. The loss of response contrast increases the amount of time required to obtain estimates with a particular confidence interval.

## Discussion

Scientists recognize the tension between using established methods and extending these methods to make new discoveries. Scientists must have confidence in the validity of the methods which advocates for a conservative approach to software use: select tools that have been widely tested within the community. But science is also about exploring new ideas and methods which advocates for a more liberal approach: try new methods that have the potential to expand our understanding. In practice, most scientists are advocates of both creativity and caution.

It is this tension that motivates us to work with the validation framework described in this paper. We find it too confining to follow the oft-repeated suggestion that investigators rely upon a few agreed upon packages. But if we are to explore, it is essential to use a testing framework that can help (a) developers to test new software, (b) research scientists to verify the software's accuracy and compare different implementations, and (c) reviewers to evaluate the methods used in publications and grants. A public validation framework, using reproducible methods and shared ground-truth dataset, is a useful compromise between caution and

creativity. In the future we plan to contribute more ground truth datasets and analyses of pRF tools, and we also hope to add validations of additional neuroimaging tools. We developed the containers so that they can accommodate other neuroimaging tools. We hope to work with others to extend the validation framework to validate tools in diffusion-weighted imaging, quantitative MRI, ECoG measures, resting-state, and anatomical methods.

Next, to illustrate the usage and utility of the framework with a practical example, we discuss the contributions of our pRF-Validation implementation.

### pRF-Validation

The pRF-Validation framework exposed meaningful implementation differences. The differences that we discovered would lead investigators to report different quantitative parameter estimates from otherwise equivalent data. This finding illustrates the value of the validation framework in making meaningful comparisons across labs. We can reduce conflicts that arise when two labs report differences that arise from the prf-Analyze tool rather than empirical measurements.

We illustrate the framework with population receptive field (pRF) tools because there are many: We found 11 public pRF implementations (https://github.com/vistalab/PRFmodel/wiki/Public-pRF-implementations). It is likely that extending the validation framework to test the additional prf-Analyze tools will reveal other meaningful differences. We hope that making the ground truth datasets and the synthesis and analysis containers available to the community will help software developers who are creating new methods, and researchers to have confidence in their tool choice.

Using pRF-Validation we made several observations that can be used to interpret the literature and inform future experimental designs. First, we discovered and reported issues in some implementations. Second, there is a significant dependence of the pRF size estimate on the HRF, and the HRF model differs significantly between implementations. This should be accounted for when comparing numerical estimates of pRF size using the different tools. We found no significant effect on position estimates of the pRF, and we described different approaches to mitigating the impact of uncertainty about the HRF using both computational and empirical means.

**Center location versus size.** We found the median polar angle and eccentricity errors are little not affected by HRF differences (see Figs 7 and 8). This might be in line with other reports on center location reliability [19–23].

**Mitigation strategies.** Validating the software using different types of stimuli clarified important issues. Randomizing the position of the stimuli is useful for mitigation of the unwanted effects of HRF variation, but it produces a lower SNR in the response compared to a stimulus that continuously sweeps through the visual field [23]. If the subject's HRF can be accurately estimated—say by an independent measurement—then the continuous sweep may be preferable. Otherwise, a randomized presentation may be preferred. Further, we learned that differences between parameter estimates from continuous and randomized presentations are to be expected based on the software methods and should not be attributed to psychological effects such as attention [23,36].

**Comparison of models.** We only used circular models to illustrate the usage of the software validation framework, but the framework can be used to compare circular to elliptical Gaussians or to Difference of Gaussians [22,24]. We can also ask what kinds of stimuli would be best-suited for distinguishing different models, similar to work in other computational domains [37]. We observe that not all packages estimate pRF-s near stimulus edges; this may not be a measurement issue, it may be a software design issue.

## Outliers

Because pRF models are not linear with respect to their parameters, the solutions are found by non-linear search algorithms. The algorithms sometimes return non-optimal local minima. It is not unusual for many of the algorithms we tested to return extreme cases that are outliers. Some of the algorithms mitigate this problem by placing a bound on the returned estimate. Other algorithms incorporate resampling, running the search with different starting points and testing for agreement between the returned parameters.

The simulations included 100 repetitions for the same voxel, with randomized noise. In these simulations the median result is close to the real solution. We assume that this approximates what is observed with real data; the median of the results yields the right solution but that there are outliers. In practice we suggest that repeating the pRF analysis with different seeds three or four times will approximate the median solution. Alternatively, it is possible to run the analysis by adding a small amount of noise to every real fMRI time series and report the median result.

There is another approach which we have not yet seen implemented. In many parts of visual cortex nearby voxels will have similar pRF parameters. When this is known to be true, it is possible to increase the effective SNR of the measurements by including an expectation that the estimates from nearby voxels will be similar.

## Guiding experimental design

We are using the tool as a simulator to design new types of stimuli and experiments prior to data acquisition. For example, here we used the tool as a simulator to identify the two empirical methods for mitigating the unwanted impact of the unknown HRF. This approach may also be useful when making a stimulus and experimental design choices. The prf-Synthesize and prf-Analyze tools are helpful for assessing whether the design will support a meaningful test of a specific hypothesis.

## Limitations

Validation testing provides an opportunity to consider several limitations of the population receptive field methods. We comment on these first, and then we make some observations about the validation framework itself.

**Noise and HRF stimulus-dependence.** We estimated the noise in the BOLD signal from an on/off experiment. We are not sure that this noise estimate is perfectly matched to the noise in the retinotopic measurement. This is true for both the signal amplitude and the correlation. In particular, the time series are synthesized independently. We expect that adjacent voxels in the BOLD measurements generally have correlated noise. Incorporating this spatial correlation may be useful. Moreover, we assumed additive noise but there may also be multiplicative noise, perhaps due to variations in attention or arousal [38]. In the future, it may be useful to express certain types of noise with respect to the input units, such as eye movements, rather than output units (% BOLD). As noted earlier, the HRF may be stimulus dependent, it may vary across the visual cortex, and it may differ significantly between participants.

**prf-Analyze parameter selection.** We made a best-faith effort to optimize the configuration parameters, sometimes consulting with the authors. Further, in creating the Docker containers we use a JSON file that enables the user to configure the prf-Analyze tools. There is room to improve the documentation of the parameters in order to help researchers use the tools more effectively. The validation framework supports the documentation because users can vary parameters and explore the consequences.

**Creating containers.** Exploring the validation framework is simple for most users: they need only a text editor and Docker. It would be desirable to make contributions to the validation framework equally simple. At present contributing a new prf-Analyze tool involves the following steps. If the pRF tool is implemented in Python, the developer must configure a Docker container that includes all of the required dependencies. If the tool is implemented in Matlab, the source code must be compiled and placed in a Docker container that includes the Matlab run-time environment (we provide this for Matlab 2018b). In both cases, the tool must read inputs and write outputs according to the BIDS standard. We provide functions for this purpose (see the prf-Validation Guide).

## Conclusion

We developed a software validation framework that is applicable in principle to many neuro-imaging software tools. By its reliance on containerization techniques and BIDS, the framework is replicable and flexible. Container technology is a flexible industry standard in that software written in almost any language can be containerized and run on any operating system. For example, the pRF methods we analyzed were written in Python, C, and Matlab and the containers were executed on several operating systems. BIDS is a helpful data-organization standard for most MRI modalities that gives inputs and outputs of the framework a clear interpretation.

Considering future development, we note that implementing the synthesis component is deeply related to creating a model from stimulus to experimental measurement. As we develop better synthesis tools, we are building a model of the system response. This validation framework, nominally designed to improve software, also advances us towards the fundamental goal of creating computable models of the experiments.

## Supporting information

**S1 File. Detailed working steps of the 3-element validation framework and Fig 7 equivalent plots with different parameter combinations.**
(DOCX)

## Author Contributions

**Conceptualization:** Garikoitz Lerma-Usabiaga, Jonathan Winawer, Brian A. Wandell.

**Data curation:** Garikoitz Lerma-Usabiaga, Noah Benson, Brian A. Wandell.

**Formal analysis:** Garikoitz Lerma-Usabiaga, Jonathan Winawer, Brian A. Wandell.

**Funding acquisition:** Garikoitz Lerma-Usabiaga, Brian A. Wandell.

**Investigation:** Garikoitz Lerma-Usabiaga, Jonathan Winawer, Brian A. Wandell.

**Methodology:** Garikoitz Lerma-Usabiaga, Noah Benson, Jonathan Winawer, Brian A. Wandell.

**Project administration:** Garikoitz Lerma-Usabiaga, Brian A. Wandell.

**Resources:** Garikoitz Lerma-Usabiaga, Jonathan Winawer, Brian A. Wandell.

**Software:** Garikoitz Lerma-Usabiaga, Noah Benson, Jonathan Winawer, Brian A. Wandell.

**Supervision:** Garikoitz Lerma-Usabiaga, Jonathan Winawer, Brian A. Wandell.

**Validation:** Garikoitz Lerma-Usabiaga, Noah Benson, Jonathan Winawer, Brian A. Wandell.

**Visualization:** Garikoitz Lerma-Usabiaga, Jonathan Winawer, Brian A. Wandell.

**Writing – original draft:** Garikoitz Lerma-Usabiaga, Jonathan Winawer, Brian A. Wandell.

**Writing – review & editing:** Garikoitz Lerma-Usabiaga, Noah Benson, Jonathan Winawer, Brian A. Wandell.

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
