## [Decision Letter · Decision Letter 0]

24 Mar 2020

Dear Mr. Lerma-Usabiaga,

Thank you very much for submitting your manuscript "A validation framework for neuroimaging software: the case of population receptive fields" for consideration at PLOS Computational Biology.

As with all papers reviewed by the journal, your manuscript was reviewed by members of the editorial board and by several independent reviewers. In light of the reviews (below this email), we would like to invite the resubmission of a significantly-revised version that takes into account the reviewers' comments.

We cannot make any decision about publication until we have seen the revised manuscript and your response to the reviewers' comments. Your revised manuscript is also likely to be sent to reviewers for further evaluation.

Sincerely,

Saad Jbabdi

Associate Editor

PLOS Computational Biology

Samuel Gershman

Deputy Editor

PLOS Computational Biology

Reviewer's Responses to Questions

**Comments to the Authors:**

Reviewer #1: This study contains a worthwhile effort to validate the estimates of population receptive field (pRF) parameters with a range of common software packages used in the literature. I welcome the authors' intention because such validation is important and desperately needed. One would certainly hope that the authors of previous pRF analysis tools already performed such simulations on their methods in-house, and in fact several authors have in fact published such validations, which would deserve some more discussion. Nevertheless, a more comprehensive effort is certainly very timely, considering the popularity of these methods now. I would be very interested to extend the authors' approach to our analysis tools (although I am not sure I have the technical knowledge to use the Docker approach presented here without further assistance). In general, I very much like the authors' suggestion of using such an approach to guide the most effective experimental design of future studies.

That said, I think there are a number of limitations to the study presented here and this could make this study more comprehensive and increase its impact on the field:

1. As far as I understand, the simulated BOLD time series are based on Kay's analyzePRF methods. As such it doesn't seem very surprising that this method seems to reproduce the ground truth most accurately in the noise-free data (Figure 5). This design seems asymmetric. If generating the pRF time course using the AFNI method, will this result in more precise estimates for AFNI?

2. The differences between the methods also deserve more discussion. To my knowledge, the AFNI method is purely based on an extensive grid search whereas in mrVista this varies between studies (although the standard method includes an optimisation step). I assume default parameters were used for each method but it would be useful to compare this, perhaps in the form of a table.

3. It is unclear to me why only one pRF at one location and size was chosen for the assessments. Surely the situation might differ considerably across different ground truth eccentricities or with different ground truth pRF sizes (which in turn would probably have implications for different visual areas)?

4. Why did you choose an oriented 2D Gaussian chosen as the underlying pRF model for the simulations? As far as I can tell, the most commonly used pRF model is a circularly symmetric 2D Gaussian. Unless I misunderstood, it also seems like the analysis of he data you present reflects a circularly symmetric model and the data analysis appears to be based on a single example pRF that is also symmetric (see point 3).

5. I may have missed this but is the data using the difference-of-Gaussians (DoG) pRF model actually used in any of the analyses? If not, then it might be sensible to remove this (you could say that this is possible to synthesise if desired). I am also not clear as to why this DoG model seems to have the same sigma (pRF size) parameters for the positive and negative components. For the DoG models implemented by Zuiderbaan et al (2012, J Vis), Schwarzkopf et al (2014, J Neurosci), or Anderson et al (2017, J Neurosci) the negative component is typically a larger surround and thus would have a larger sigma than in positive, central component (and these models did not use oriented, non-circular pRFs).

6. The analysis about sweep duration and stimulus randomisation doesn't appear until the Discussion. I feel this belongs into the results section. It makes sense to me that the randomised designs have a weaker signal-to-noise ratio and this matches what Binda et al (2013, J Vis) observed but then again we found in Infanti & Schwarzkopf (2020, NeuroImage) that randomisation did not impact signal-to-noise all that much - although this could represent a ceiling effect because we only estimated polar angles in those experiments.

7. You suggest that polar angle and eccentricity errors between the methods are negligible. This certainly matches the findings based on the reliability from Senden et al (2014, PLoS One), van Dijk et al (2016, NeuroImage), and Benson et al (2018, J Vis). I think these findings should be discussed in the context of these reliability studies. Of course, the aforementioned caveat applies that all the present results only speak to one simulated pRF of with x,y position 3,3 and radius 2 deg. It would be important to quantify this across a plausible range of pRF parameters.

Minor points:

Line 140: "is comprised of" - I'm probably pedantic and also realise that this is a grey area - but I'd change this to "comprises" or "consists of"

Line 145: typo - "noise-free" should be "synthetic"

Line 261: "summer 2019" - while I know what you mean this is ambiguous unless you're a Flat Earther and don't believe in the existence of the southern hemisphere. Change to "mid-2019"?

Line 284: In all the figure captions you state that the radius SD of the simulated pRF was 2 deg. This matches also the ground truth shown in the figures themselves. But here in the text there appears to be a typo as it is reported as 1 deg.

Jargon and figure labels: these could be made more consistent. For example you seem to move between signal percent change, percent signal change, and relative amplitude - sometimes even within the same figure.

Citations for pRF models: You should probably cite Zuiderbaan et al (2012, J Vis) for the DoG pRF models (although this is a simpler and somewhat different model than you use here - see point 5). For the oriented pRF model you should probably cite Silson et al (2018, J Neurosci). As mentioned earlier, I think the most widely used model is the circularly symmetric Gaussian model first presented by Dumoulin & Wandell (2008).

Reference list citation 28: My name is misspelled in that reference as "Sam Schwarzkopf D" when the correct reference should read "Schwarzkopf DS". I've noticed this before and may have something to do with how reference managers parse the author list. Sorry...

Sam Schwarzkopf

University of Auckland

Reviewer #2: This paper presents a general computational framework for evaluating neuroimaging software, with both computational reproducibility and validity.

The authors demonstrate this framework by applying it to population receptive field (pRF) methods for fMRI, and they present initial findings comparing different pRF tools using this framework.

Two highlighted features of the framework are the containerised architecture (with BIDS implementation) which ensures reproducibility, and the generation of synthetic data with known ground truth to evaluate validity.

Within the pRF context, the authors report that this validation framework allowed them to identify and report issues with some pRF tools, and a significant dependence on the HRF model used in different implementations.

Furthermore, it is proposed that the framework can be extended to evaluate other neuroimaging algorithms.

I acknowledge the importance and need for such frameworks and their utility for comparing across different neuroimaging tools within a common validation framework.

The paper is clear, and mostly well written. However, I am confused about the emphasis of this paper. Initially, from the title, abstract and introduction, I was under the impression that this paper was describing a general tool that could be applied to many neuroimaging contexts, but which would be demonstrated in the pRF context. However, through the methods results and discussion, the focus seem to shift heavily towards the pRF specific application.

I am left unclear if this is a general framework that is demonstrated in a pRF application, or a pRF framework that could, but has not been, adapted to be more general…. I appreciate that this may be a subtle difference, but to many readers it will be an important distinction.

If it is the former, then much more detail is required about how the framework would need to be adapted for other neuroimaging contexts, and how much effort that would be.

If the latter, then I suggest you embrace this as a pRF validation tool and downplay the more general sounding descriptions in the earlier parts of the paper, particularly the abstract and title. You should also promote the pRF results, which figure prominently in the discussion, to the abstract. In this form, you will need to take the evaluation of the pRF tools to its full conclusion; as you currently only describe initial findings.

Major issues:

Other than the emphasis, which I have already discussed, I have no other major issues with this paper.

Minor issues:

I think that the limitations section could benefit from some discussion of ground-truth versus truth, and generalisation. As this framework is dependent on synthetic data, the risk is that software will be optimised towards the ground-truth of synthetic data, which could reduce its generalisation to “real” data. Perhaps it would have been worthwhile performing evaluation with real data alongside the synthesised data, especially considering that the x-Analyze model already supports this.

**Have all data underlying the figures and results presented in the manuscript been provided?**

Reviewer #1: Yes

Reviewer #2: Yes

PLOS authors have the option to publish the peer review history of their article (what does this mean?). If published, this will include your full peer review and any attached files.

Reviewer #1: Yes: Sam Schwarzkopf

Reviewer #2: No
---

## [Decision Letter · Decision Letter 1]

3 May 2020

Dear Mr. Lerma-Usabiaga,

We are pleased to inform you that your manuscript 'A validation framework for neuroimaging software: the case of population receptive fields' has been provisionally accepted for publication in PLOS Computational Biology.

Best regards,

Saad Jbabdi

Associate Editor

PLOS Computational Biology

Samuel Gershman

Deputy Editor

PLOS Computational Biology

Reviewer's Responses to Questions

**Comments to the Authors:**

Reviewer #1: The authors have addressed my previous comments. With regard to previous literature on validation, I was really referring to previous work that used simulated time courses for assessing the validity of pRF model fits, similar to what the authors do here. Such approaches were for example used by Binda et al 2013 or Senden et al 2014. While the motivation behind that work may have more to do with the scientific research questions of these respective studies, it is arguably also a form of software validation. But perhaps discussing this is outside the scope of the present study.

I am sorry if I have missed this, but could you please give a little more detail as to how the slowed down stimulus sequences worked? Were they slowed down by longer duration at each bar step or by having more bar steps at 1 second per step?

Sam Schwarzkopf

University of Auckland

Reviewer #2: I am satisfied that the authors have made a genuine attempt to address my concerns. There is now a clearer distinction between the general applicability of the tool, and the specific application to pRF. I still would prefer a clearer title, but I am happy to leave that to the authors discretion.

**Have all data underlying the figures and results presented in the manuscript been provided?**

Reviewer #1: Yes

Reviewer #2: Yes

PLOS authors have the option to publish the peer review history of their article (what does this mean?). If published, this will include your full peer review and any attached files.

Reviewer #1: No

Reviewer #2: No

---

## [Editor Report · Acceptance letter]

16 Jun 2020

PCOMPBIOL-D-20-00318R1 

A validation framework for neuroimaging software: the case of population receptive fields

Dear Dr Lerma-Usabiaga,

I am pleased to inform you that your manuscript has been formally accepted for publication in PLOS Computational Biology. Your manuscript is now with our production department and you will be notified of the publication date in due course.

With kind regards,

Laura Mallard
